# Estimated reduction in obesity prevalence and costs of a 20% and 30% ad valorem excise tax to sugar-sweetened beverages in Brazil: A modeling study

**Ana Basto-Abreu**[1], **Rossana Torres-Alvarez**[2,3], **Tonatiuh Barrientos-Gutierrez**[1]*, **Paula Pereda**[4,5], **Ana Clara Duran**[6]*

1 Center for Population Health Research, National Institute of Public Health, Cuernavaca, Mexico, 2 Department of Integrative Oncology, BC Cancer Research Institute, Vancouver, Canada, 3 School of Population and Public Health, University of British Columbia, Vancouver, Canada, 4 Department of Economics, School of Economics and Business, University of São Paulo, São Paulo, Brazil, 5 Center for Epidemiological Studies in Nutrition and Health, University of São Paulo, São Paulo, Brazil, 6 Center for Food Studies and Research, University of Campinas; Center for Epidemiological Studies in Nutrition and Health, University of São Paulo, São Paulo, Brazil

* tbarrientos@insp.mx (TB-G); anaduran@unicamp.br (ACD)

**Data Availability Statement:** The script and database for this analysis can be found in the

## Abstract

### Background

The consumption of sugar-sweetened beverages (SSBs) is associated with obesity, metabolic diseases, and incremental healthcare costs. Given their health consequences, the World Health Organization (WHO) recommended that countries implement taxes on SSB. Over the last 10 years, obesity prevalence has almost doubled in Brazil, yet, in 2016, the Brazilian government cut the existing federal SSB taxes to their current 4%. Since 2022, a bill to impose a 20% tax on SSB has been under discussion in the Brazilian Senate. To simulate the potential impact of increasing taxes on SSB in Brazil, we aimed to estimate the price-elasticity of SSB and the potential impact of a new 20% or 30% excise SSB tax on consumption, obesity prevalence, and cost savings.

### Methods and findings

Using household purchases data from the Brazilian Household Budget Survey (POF) from 2017/2018, we estimated constant elasticity regressions. We used a log-log specification by income level for all beverage categories: (1) sugar-sweetened beverages; (2) alcoholic beverages; (3) unsweetened beverages; and (4) low-calorie or artificially sweetened beverages. We estimated the adult nationwide baseline intake for each beverage category using 24-h dietary recall data collected in 2017/2018. Taking group one as the taxed beverages, we applied the price and cross-price elasticities to the baseline intake data, we obtained changes in caloric intake. The caloric reduction was introduced into an individual dynamic model to estimate changes in weight and obesity prevalence. No benefits on cost savings were modeled during the first 3 years of intervention to account for the time lag in obesity

following repository: https://github.com/INSP-RH/Brazil_SSB_Tax.

**Funding:** This work was funded by a Bloomberg Philanthropies grant (5104695 to ACD; https://www.bloomberg.org/). The funders had no role in study design, data collection and analysis, decision to publish, or preparation of the manuscript.

**Competing interests:** The authors have declared that no competing interests exist.

**Abbreviations:** BMI, body mass index; SSB, sugar-sweetened beverages; WHO, World Health Organization.

cases to reduce costs. We multiplied the reduction in obesity cases during 7 years by the obesity costs per capita to predict the costs savings attributable to the sweetened beverage tax. SSB price elasticities were higher among the lowest tertile of income (−1.24) than in the highest income tertile (−1.13), and cross-price elasticities suggest SSB were weakly substituted by milk, water, and 100% fruit juices. We estimated a caloric change of −17.3 kcal/day/person under a 20% excise tax and −25.9 kcal/day/person under a 30% tax. Ten years after implementation, a 20% tax is expected to reduce obesity prevalence by 6.7%; 9.1% for a 30% tax. These reductions translate into a −2.8 million and −3.8 million obesity cases for a 20% and 30% tax, respectively, and a reduction of $US 13.3 billion and $US 17.9 billion in obesity costs over 10 years for a 20% and 30% tax, respectively. Study limitations include using a quantile distribution method to adjust self-reported baseline weight and height, which could be insufficient to correct for reporting bias; also, weight, height, and physical activity were assumed to be steady over time.

## Conclusions

Adding a 20% to 30% excise tax on top of Brazil's current federal tax could help to reduce the consumption of ultra-processed beverages, empty calories, and body weight while avoiding large health-related costs. Given the recent cuts to SSB taxes in Brazil, a program to revise and implement excise taxes could prove beneficial for the Brazilian population.

## Author summary

### Why was this study done?

- The consumption of sugar-sweetened beverages (SSBs) is associated with obesity, metabolic diseases, and incremental healthcare costs.
- The World Health Organization (WHO) recommends governments to tax SSB to improve diet and reduce chronic diseases.
- Since 2022, a bill to impose a 20% tax on SSB is under discussion in the Brazilian Senate.

### What did the researchers do and find?

- We estimated SSB own- and cross-price elasticities by socioeconomic status and simulated the potential effects of introducing a 20% excise tax to SSB in consumption. If a 20% tax is introduced, we would expect a 16.9 kcal/day/person caloric reduction among Brazilian adults.
- We used an individual dynamic weight change model to translate caloric reductions into obesity reductions. We estimated an expected 6.7% reduction in obesity, which could lead to USD 13.3 billion healthcare cost savings over 10 years.
- Larger benefits of the tax are expected to be experienced by younger adults and by people from high-income groups.

**What do these findings mean?**

- The implementation of the SSB tax is expected to produce important reductions in population weight and obesity prevalence, while saving health-related costs.

- Limitations of our analysis include assuming no change in weight under the no-intervention scenario and using a quantile distribution method to adjust self-reported baseline weight and height.

## Introduction

Consumption of sugar-sweetened beverages (SSBs) is associated with increased caloric intake, weight gain, and the development of multiple chronic diseases, such as diabetes, metabolic syndrome, and cancer [1–3]. Governments worldwide are establishing structural interventions to improve consumers' choices. Following the experience of alcohol and tobacco, taxes on SSB have gained international attention as an effective measure to reduce sugar consumption and improve health [4]. A systematic review analyzed the impact of SSB taxes in the United States, finding on average that a 10% tax on SSBs produces a 10% decrease in consumption [5]. Recently, the World Health Organization (WHO) recommended that countries implement a 20% tax on sugary beverages to improve diet and reduce chronic diseases [6].

Latin America has been at the forefront of SSB taxes [7]. In 2014, Mexico implemented a 10% SSB tax that reduced consumption by 9.7% 2 years after implementation [8]. The 10% SSB tax in Mexico was simulated over 10 years, predicting a 2.5% reduction in the prevalence of obesity among adults [9]. Further simulation studies estimated important reductions in cardiovascular disease [10], child body weight [10,11], and obesity-related cancers [12]. In 2016, Chile implemented an integral package of interventions to reduce SSB consumption, including increasing an existing 13% tax to 18% [13,14]. By 2017, the consumption of taxed beverages decreased by 23.4%, while caloric content decreased by 27.5% [14]. SSB taxes and regulations in the Latin American region have proven to reduce consumption and are expected to produce important health benefits.

In 2010, Brazil's SSB per capita consumption was estimated at 142 ml/day/person [15]. Between 2009 and 2019, the prevalence of obesity in the country increased from 11.8% to 20.3% across all ages, but particularly among young adults [16], where SSB consumption is more frequent [15]. The rise in obesity has been concurrent with the increase in SSB consumption [17]. Despite having high consumption levels, Brazil reduced SSB federal taxes in 2016 by 23 percentage points, and the current SSB federal tax in the country is 4% [18]. Since 2022, a bill to impose a 20% tax on SSB is under discussion in the Brazilian Senate [19].

We aimed to estimate the potential impact of an SSB tax in Brazil to disincentivize SSB consumption. First, we estimated own- and cross-price elasticities for SSB and other beverages assuming a new 20% and 30% excise taxes were added to the existing federal tax. Then, we used a dynamic simulation model to estimate the expected impact of the taxes on body weight in the adult Brazilian population. Prevented obesity cases were then linked to obesity costs to estimate the obesity costs averted by such intervention over a 10-year period.

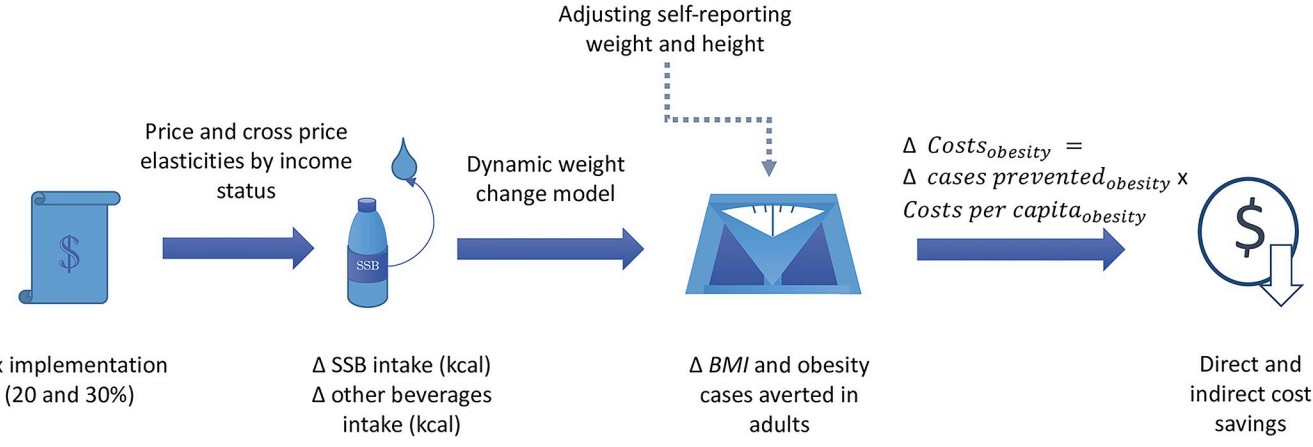

**Fig 1. Representation of the simulation strategy** [20–24]. BMI, body mass index; SSB, sugar-sweetened beverages.

## Methods

### Model overview

We used a simulation model to estimate the potential impact of an SSB tax on obesity and obesity costs in Brazil (Fig 1) as the country is discussing its implementation. First, we used purchases data to estimate own- and cross-price elasticities for SSB in Brazil; we assumed that the expected reduction in purchases is translated into a proportional reduction in consumed calories. Using a dynamic weight change model, we estimated changes in body mass index (BMI) and obesity prevalence over 10 years, then linked them to per capita obesity costs to estimate the expected obesity costs averted. We simulated 2 tax scenarios—a 20% and a 30% SSB tax—, and we considered changes in SSB intake only and the potential caloric substitution by other beverages based on the cross-price elasticities. All steps are further explained in the next subsections.

### Price elasticities by income level

To estimate the price elasticities of SSB by income level, we used the Brazilian Household Budget Survey (*Pesquisa de Orçamentos Familiares—POF*) from 2017 to 2018 (POF 2017–2018) collected by the Brazilian Institute of Geography and Statistics (IBGE) [20]. POF 2017–2018 is representative of Brazil, its states, and metropolitan regions. Within POF, we used the purchase database, which registers all food purchases of a sample of 57,920 Brazilian households within a week (representing 59,783,430 Brazilian households). We considered POF survey sampling weights in all estimates.

POF data include food acquisition—in total quantity (in kg)—and monetary value (in Brazilian reais). We divided beverages into 4 categories: (1) sugar-sweetened beverages; (2) alcoholic beverages; (3) unsweetened beverages; and (4) low-calorie or artificially sweetened beverages (light/diet). Table A in S1 Appendix describes the beverage items included in each category.

We also categorized food items to include food prices as controls in the demand regressions. Other food groups included: (i) fruits; (ii) *in natura* food; (iii) *in natura* meat; (iv) snacks; (v) sugar; (vi) other processed food; and (vii) other food items. Prices were calculated as unit values, as total expenditure divided by total quantity consumed (implicit price). The economic theory posits that zero consumption represents a choice made by consumers who assess prices and opt not to make a purchase. If a household reported zero consumption of an

item, we approximated its price by the median of the neighboring regions, under the assumption that those areas generally experience similar market prices. We estimated basic constant elasticity regressions using a log-log specification for all beverage categories, including income and squared income as covariates ($inc+inc^2$):

$$log(q_{i,b}) = \alpha_{0,b} + \sum_{f=1}^{11} \beta_{f,b} \log(p_f) + \eta_1 inc + \eta_2 inc^2 + X_i\theta_b + \varepsilon_{i,b}.$$

In which $q_{i,b}$ is the total quantity (in kg) of beverage $b$ acquired by household $i$. We included the price logarithm for all 11 categories of food and beverages (indexed by $f$). We also included several sociodemographic variables as control variables (matrix $X$), such as (i) household head (gender, years of schooling, race, and marital status), and household characteristics (total members of the household, total children within the household, and logarithm of the income in levels and quadratic).

We considered different price elasticities by income level. Income levels were categorized using the tertiles ($T$) of the distribution of income per capita of the households ($TertIncome_T$) calculated using sampling weights. Thus, the basic regression was modified according to the following expression:

$$log(q_{i,b}) = \alpha_{0,b} + \sum_{f=1}^{11}\sum_{T=1}^{3} \beta_{f,b} \log(p_f) \times TertIncome_{i,T} + X_i\theta_b + \eta_1 inc + \eta_2 inc^2 + \varepsilon_{i,b}.$$

## Baseline intake of beverages

Baseline intake of beverages and self-reported anthropometric data were collected using the POF 2017–2018. Food consumption was assessed using a single 24-h food recall, because two-day information was collected in a smaller subsample (31,762) for which sampling weights were not available. We selected individuals 20 years of age and older (37,689 adults), representing 147,852,423 individuals in the adult population in Brazil. Each subject reported the total number of beverages they consumed in standard servings, and then we transformed it to milliliters. We used the same beverages categorization as in the purchases data (Table A in S1 Appendix).

## Reduction in energy intake

To estimate the expected impact of the tax on calories, we created 2 scenarios considering: (1) the own-price elasticity for SSB by income level; and (2) both own- and cross-price elasticities for alcoholic beverages, light/diet and low-calorie or artificially sweetened beverages, and other beverages by income level. We assumed that changes in household purchases derived from price elasticities translated directly into household consumption changes. Prior studies suggest the effects of soda taxes on consumption are linear; thus, we assumed linearity in the changes in consumption for the 20% and 30% tax scenarios [25].

## Adjustment of self-reporting bias on weight and height

We adjusted self-reported height and weight from POF 2017–2018, linked to SSB consumption data, using measured data from the Brazilian National Health Survey (*Pesquisa Nacional de Saúde*–PNS 2019) [21]. The PNS 2019 is a population-based survey, nationally representative, aiming at estimating living and health conditions. This survey measured weight and height using scales, portable stadiometers, and anthropometric tapes with the appropriate training and supervisors. More information about the PNS 2019 can be found elsewhere [26]. Being

both POF and PNS nationally representative surveys, we could adjust for self-reporting bias. We adjusted self-reported weight and height from POF to match the distribution of weight and height measured in PNS 2019, following previously published methods [23]. Briefly, we calculated the difference between weight and height data quantiles from both surveys and fitted a cubic spline to smoothly construct a distribution of bias for weight and height. Then, using the quantiles of self-reported weight and height, we estimated the bias using fitted cubic splines (Figs A and B in S1 Appendix). Finally, we added the predicted difference to the self-reported data by sex, as previous articles reported a larger underestimation of weight and overestimation of height in women compared to men [27]. Details are available in Section 2 from S1 Appendix.

## Reduction in body weight, BMI, and obesity prevalence

To simulate weight changes attributable to the SSB tax on the Brazilian population, we used a microsimulation approach, a dynamic model developed by Hall and colleagues [22]. The model is implemented in the bw package in R (https://rdrr.io/github/INSP-RH/bw/). It considers the dynamic physiological adaptations to body weight that lead to changes in resting metabolic rate and physical activity [22] and has been previously implemented to estimate the impact of the sugar-sweetened beverages tax [9,28,29]. The model considers changes in extracellular fluid, glycogen, and fat and lean tissues caused by the change in caloric consumption, keeping the physical activity constant. Thus, the time lag that takes the caloric changes to be reflected in weight is considered by the model. Bodyweight is the result of the sum of fat mass and fat-free mass, and is determined using a system of ordinary differential equations. To initialize the model and obtain resting metabolic rate, we used sex, age, weight, and height from POF 2017–2018 assuming all adults were sedentary (physical activity level = 1.5). Table J and Section 3 in S1 Appendix presents more details of the simulation model and its parameters.

## Uncertainty

Obesity prevalence considered the uncertainty of POF 2017–2018.

## Sensitivity analysis: Measured weight and height from PNS 2019

We relied on self-reported weight and height to generate the expected impact of the SSB tax, adjusting the self-report bias expected in POF 2017/2018 using quantiles of weight and height data from PNS 2019. However, recent concerns have been raised about the potential limitations of this method [30]. To validate the results of our analysis, we followed an independent estimation procedure, using measured weight and height from PNS 2019 and SSB consumption from POF 2017/2018. Since these 2 surveys cannot be linked at the individual level, we estimated the average SSB consumption and BMI for 14 groups by age (20 to 39, 40 to 59, 60+) and socioeconomic status (low, middle, and high) for males and females. Then, we implemented Hall's dynamic model using the averages, generating the expected weight reduction in kilograms for each group and compared against the estimated produced using the corrected self-reported data. Results and methods for this analysis are fully available in Section 6 in S1 Appendix.

## Reduction in cases with obesity

We translated the reductions in the prevalence of obesity into obesity cases averted over 10 years using the baseline prevalence of obesity in Brazil (assuming a steady state) and the

expected reductions in obesity, multiplied by the population projections from the Brazilian Institute of Geography and Statistics for the adult population from 2021 to 2030.

## Reduction in healthcare costs

Obesity costs were obtained from a 2021 publication which estimated direct and indirect costs attributable to overweight and obesity for 8 countries, including Brazil [24]. The total cost of obesity in Brazil was estimated to be US$ 38.76 billion per year (16.19 direct and 22.57 indirect costs), using a societal perspective. Direct costs were estimated as the obesity-attributable fraction of healthcare expenditures and indirect costs included economic loss from premature mortality, missed days of work (absenteeism), and reduced productivity while at work (presenteeism). Using a systematic review that estimated that 87% of the overweight and obesity costs were attributed to obesity, we estimated the obesity costs in Brazil to be US$33.72 billion [31]. In 2018, we estimated a 25.2% obesity prevalence in the adult population in Brazil using POF, resulting in 37.2 million persons with obesity. Dividing the annual cost of obesity by the total number of obesity cases, we estimated an annual cost per obesity case of US$ 906.92. Using the Consumer Price Index Inflation rate, we converted costs from 2019 to costs in 2021 and obtained an annual cost of US$ 942.5. To account for the time lag in morbidity and mortality changes following obesity changes, no cost benefits were estimated during the first 3 years of intervention, based on average time lags for most obesity-related diseases in a dietary multi-stage life table [32]. The benefits of the full effect of the intervention on direct and indirect costs were modeled in years 4 to 10. To estimate the total cost savings, we multiplied the absolute yearly reduction in obesity prevalence by the direct and indirect obesity costs. We used a 5% discount rate to reduce overestimation [33]. A detailed description of cost estimations is included in Section 5 from S1 Appendix.

## Results

Table 1 presents the estimated price elasticities for SSB by income level. Low-income households were, on average, more sensitive to changes in SSB prices, as they had higher price

**Table 1. Price elasticities of beverages by income groups, 2018.**

| | | Sugar sweetened beverages | Unsweetened beverages | Alcoholic beverages | Light/diet beverages† |
|---|---|---|---|---|---|
| Dependent variable: Log of Quantity Acquired (in Liters) | | | | | |
| Low income | Elasticities | -1.241 | 0.046 | -0.122 | -0.174 |
| | s.e. | (0.015) | (0.017) | (0.053) | (0.124) |
| | p-value | <0.001 | 0.006 | 0.022 | 0.160 |
| Middle income | Elasticities | -1.186 | 0.030 | -0.076 | 0.094 |
| | s.e. | (0.019) | (0.021) | (0.047) | (0.110) |
| | p-value | <0.001 | 0.150 | 0.108 | 0.397 |
| High income | Elasticities | -1.126 | 0.066 | -0.141 | 0.201 |
| | s.e. | (0.022) | (0.025) | (0.048) | (0.084) |
| | p-value | <0.001 | 0.007 | 0.003 | 0.018 |
| Observations (n) | | 18,876 | 27,488 | 4,263 | 383 |

† This category includes low-calorie or artificially sweetened beverages.

All regressions used microdata from POF 2017–2018. Income levels were divided using the tertiles of the distribution of the income per capita of the households. Other controls include logarithm of prices of other products (11 categories included), of income (in levels and squared), characteristics of household head (schooling, age, gender, race), total number of household members, and total number of children living within the household (all ages). All regressions are weighted using sample survey weights. Standard errors (se) are clustered by socioeconomic status and presented in parentheses.

**Table 2. Baseline caloric intake from SSB and other beverages (alcoholic, diet/light, and unsweetened beverages).**

| | Population (million) | Baseline intake | | Caloric changes from SSB | | | | Caloric changes from other beverages | | | |
|---|---|---|---|---|---|---|---|---|---|---|---|
| | | | | 20% tax | | 30% tax | | 20% tax | | 30% tax | |
| | | kcal/ day | 95% CI* | kcal/ day | 95% CI* | kcal/ day | 95% CI* | kcal/ day | 95% CI* | kcal/ day | 95% CI* |
| Total | 147.9 | 71.8 | 69.3, 74.4 | −16.9 | −17.5, −16.3 | −25.3 | −26.2, −24.4 | −0.4 | −0.4, −0.3 | −0.6 | −0.7, −0.5 |
| **Sex** | | | | | | | | | | | |
| *Female* | 78.2 | 65.1 | 62.4, 67.7 | −15.3 | −15.9, −14.7 | −22.9 | −23.9, −22.0 | −0.1 | −0.1, −0.0 | −0.1 | −0.2, −0.0 |
| *Male* | 69.6 | 79.5 | 75.8, 83.1 | −18.6 | −19.5, −17.8 | −27.9 | −29.2, −26.7 | −0.7 | −0.8, −0.6 | −1.0 | −1.2, −0.9 |
| **Age groups** | | | | | | | | | | | |
| *20–39* | 62.2 | 92.2 | 88.0, 96.5 | −21.7 | −22.7, −20.7 | −32.6 | −34.1, −31.1 | −0.4 | −0.5, −0.3 | −0.7 | −0.8, −0.5 |
| *40–59* | 53.9 | 62.9 | 59.4, 66.4 | −14.7 | −15.5, −13.9 | −22.1 | −23.3, −20.8 | −0.5 | −0.6, −0.3 | −0.7 | −0.8, −0.5 |
| *60+* | 31.7 | 46.9 | 43.3, 50.6 | −11.0 | −11.9, −10.2 | −16.6 | −17.8, −15.3 | −0.1 | −0.2, −0.0 | −0.2 | −0.3, −0.1 |
| **Income status** | | | | | | | | | | | |
| *Low* | 47.4 | 53.5 | 49.9, 57.1 | −13.3 | −14.2, −12.4 | −19.9 | −21.3, −18.6 | −0.2 | −0.3, −0.2 | −0.4 | −0.5, −0.2 |
| *Middle* | 49.1 | 74.0 | 70.2, 77.8 | −17.5 | −18.5, −16.6 | −26.3 | −27.7, −25.0 | −0.2 | −0.3, −0.2 | −0.3 | −0.4, −0.2 |
| *High* | 51.4 | 86.7 | 81.7, 91.8 | −19.5 | −20.7, −18.4 | −29.3 | −31.0, −27.6 | −0.6 | −0.8, −0.5 | −0.9 | −1.2, −0.7 |

* Confidence intervals were generated based on the uncertainty of *POF 2017–2018*.

elasticities than higher-income households. SSB price elasticities ranged from −1.13 (high income) to −1.24 (low income). Substitution effects from increases in SSB prices were only observed for unsweetened beverages (which included milk, water, and 100% fruit juices) in low- and high-income households and for light/diet beverages/low-calorie or artificially sweetened beverages for high-income households. We observed a complementary behavior between the acquisition of SSB and alcoholic beverages (if one product decreased, the other also decreased).

Table 2 presents the baseline intake of SSB and other beverages among Brazilian adults in 2018. We estimated a baseline intake of 72 kcal/person/day from SSB. The expected caloric change after the SSB tax was estimated to be −16.9 kcal/person/day for 20% and −25.3 kcal/day for 30% tax. If we included changes in other beverages to allow for substitution, expected caloric changes increased to −17.3 kcal/person/day for 20% (an extra reduction of 0.4 kcal) and −25.9 kcal/person/day for 30% tax (an extra reduction of 0.6 kcal). Expected caloric changes should be higher among younger than older adults and higher income levels. The caloric change was translated into −0.75 kg in females and −0.83 kg in males. In our sensitivity analysis using measured weight and height and Hall's model at aggregated level, females were estimated to reduce 0.78 kg and males 0.84 kg (Table H in S1 appendix).

Table 3 presents the expected impact of the SSB tax over the obesity prevalence, 10 years after implementation, using own- and cross-price elasticities. The obesity prevalence is expected to decrease from 25.2% to 23.6%, resulting in a 6.3% reduction with a 20% tax and an 8.7% obesity reduction with a 30% tax. This reduction is expected to be higher among younger (−8.3%) than older adults (−3.6%) and among higher (−7.8%) than lower-income levels (−5.4%). Allowing for substitution using cross-price elasticities, the expected decrease in the obesity prevalence becomes larger, being −6.7% for 20% and −9.1% for a 30% tax.

Fig 2 shows the expected obesity reduction in cases and potential cost savings 10 years after implementing the SSB tax, using own-price elasticities. A 20% tax is expected to reduce the number of people living with obesity by 2.8 million and by 3.7 million with a 30% tax; this would translate into cost savings of $US 13.1 billion with a 20% tax and $US 17.6 billion with a 30% tax. Using cross-price elasticities, the reduction in the number of people living in obesity

**Table 3. Baseline obesity in Brazil and expected obesity reduction after the implementation of a tax on SSBs, considering caloric changes from sugar-sweetened beverages and from all beverages.**

| | Baseline obesity | | Obesity changes from SSB | | | | | | Obesity changes from all beverages | | | | | |
|---|---|---|---|---|---|---|---|---|---|---|---|---|---|---|
| | | | 20% tax | | | 30% tax | | | 20% tax | | | 30% tax | | |
| | | | Absolute change | | Relative change | Absolute change | | Relative change | Absolute change | | Relative change | Absolute change | | Relative change |
| | % | 95% CI* | pp | 95% CI* | (%) | pp | 95% CI* | (%) | pp | 95% CI* | (%) | pp | 95% CI* | (%) |
| Total | 25.2 | 24.4, 26.0 | −1.6 | −1.9, −1.4 | −6.3% | −2.2 | −2.5, −2.0 | −8.7% | −1.7 | −1.9, −1.4 | −6.7% | −2.3 | −2.5, 2.0 | −9.1% |
| **Sex** | | | | | | | | | | | | | | |
| *Female* | 28.8 | 27.8, 29.8 | −1.4 | −1.8, −1.2 | −4.9% | −2.1 | −2.5, −1.7 | −7.3% | −1.5 | −1.8, −1.2 | −5.2% | −2.1 | −2.5, 1.8 | −7.3% |
| *Male* | 21.2 | 20.2, 22.2 | −1.9 | −2.3, −1.5 | −9.0% | −2.4 | −2.8, −2.0 | −11.3% | −1.9 | −2.3, −1.6 | −9.0% | −2.4 | −2.9, −2.1 | −11.3% |
| **Age groups** | | | | | | | | | | | | | | |
| *20–39* | 21.8 | 20.7, 23.0 | −1.8 | −2.2, −1.5 | −8.3% | −2.4 | −2.9, 2.0 | −11.0% | −1.8 | −2.3, −1.5 | −8.3% | −2.5 | −3.0, −2.1 | −11.5% |
| *40–59* | 29.3 | 28.1, 30.5 | −1.9 | −2.4, −1.5 | −6.5% | −2.4 | −2.9, −2.0 | −8.2% | −2.0 | −2.5, −1.6 | −6.8% | −2.4 | −2.9, −2.0 | −8.2% |
| *60+* | 24.8 | 23.4, 26.4 | −0.9 | −1.4, −0.5 | −3.6% | −1.5 | −2.2, −1.0 | −6.0% | −0.9 | −1.4, −0.6 | −3.6% | −1.5 | −2.2, −1.0 | −6.0% |
| **Income status** | | | | | | | | | | | | | | |
| *Low* | 22.3 | 21.1, 23.4 | −1.2 | -1.6, −0.9 | −5.4% | −1.7 | −2.2, -1.3 | −7.6% | −1.2 | −1.7, −0.9 | −5.4% | −1.7 | −2.2, −1.3 | −7.6% |
| *Middle* | 26.2 | 25.0, 27.4 | −1.6 | −2.0, −1.3 | −6.1% | −2.2 | −2.7, −1.9 | −8.4% | −1.7 | −2.1, −1.3 | −6.5% | −2.3 | −2.8, −1.9 | −8.8% |
| *High* | 27.0 | 25.5, 28.5 | −2.1 | −2.7, −1.6 | −7.8% | −2.7 | −3.3, −2.2 | −10.0% | −2.1 | −2.7, −1.6 | −7.8% | −2.7 | −3.3, −2.2 | −10.0% |

* Confidence intervals were generated based on the uncertainty of *POF 2017–2018*.

is similar with −2.8 and −3.8 million for a 20% and 30% tax, respectively; cost savings reaches $US 13.3 and $US 17.9 billion with a 20% and 30% tax, respectively.

## Discussion

We aimed to estimate the own and cross-price elasticities of SSB and translate them into the expected impact of a 20% and a 30% SSB tax on obesity over 10 years in Brazil. In Brazil, we found that SSB is elastic and weakly substituted for milk, water, 100% fruit juices, and low-calorie or artificially sweetened beverages, such as diet/light beverages. Low-income households were more sensitive to price increases and had a higher elasticity than other income levels. Considering own- and cross-price elasticities, we estimated that a 20% tax on SSB could translate into a 6.7% reduction in the obesity prevalence and US$ 13.3 billion savings in obesity costs over 10 years after implementation. A 30% tax was estimated to reduce 9.1% obesity prevalence and US$ 17.9 billion in the obesity costs over 10 years. The largest benefits of the tax are expected to be experienced by males, younger adults and by people in the high-income group.

Estimating the impact of price increases on demand for SSB and other beverages in Brazil is key to assess the potential impact of a SSB tax. Our own price-elasticities indicates that a 10% price increase to SSB in Brazil should result in a 12.4% reduction in purchases among households in lowest income group, 11.9% in households in the second income group, and 11.3% in the top income group. This is lower to what has been previously estimated in Brazil, where a 10% price increase was expected to produce a 13.6% reduction in carbonated SSB, but higher than a meta-analysis of observational studies in which a 10% tax was associated to a 10% reduction in SSB purchases and intake [5,34]. Our analysis suggests that SSB purchases in Brazil will decrease if prices increase, which indicate that an excise tax on SSB would contribute to lower the consumption of SSB among the Brazilian adult population.

We analyzed cross-price elasticities, finding only a weak substitution for milk, water, and 100% fruit juices. Substitution for these beverages was previously reported in a meta-analysis that included US, Mexico, France, and Brazil studies [35]. In contrast, our findings suggest

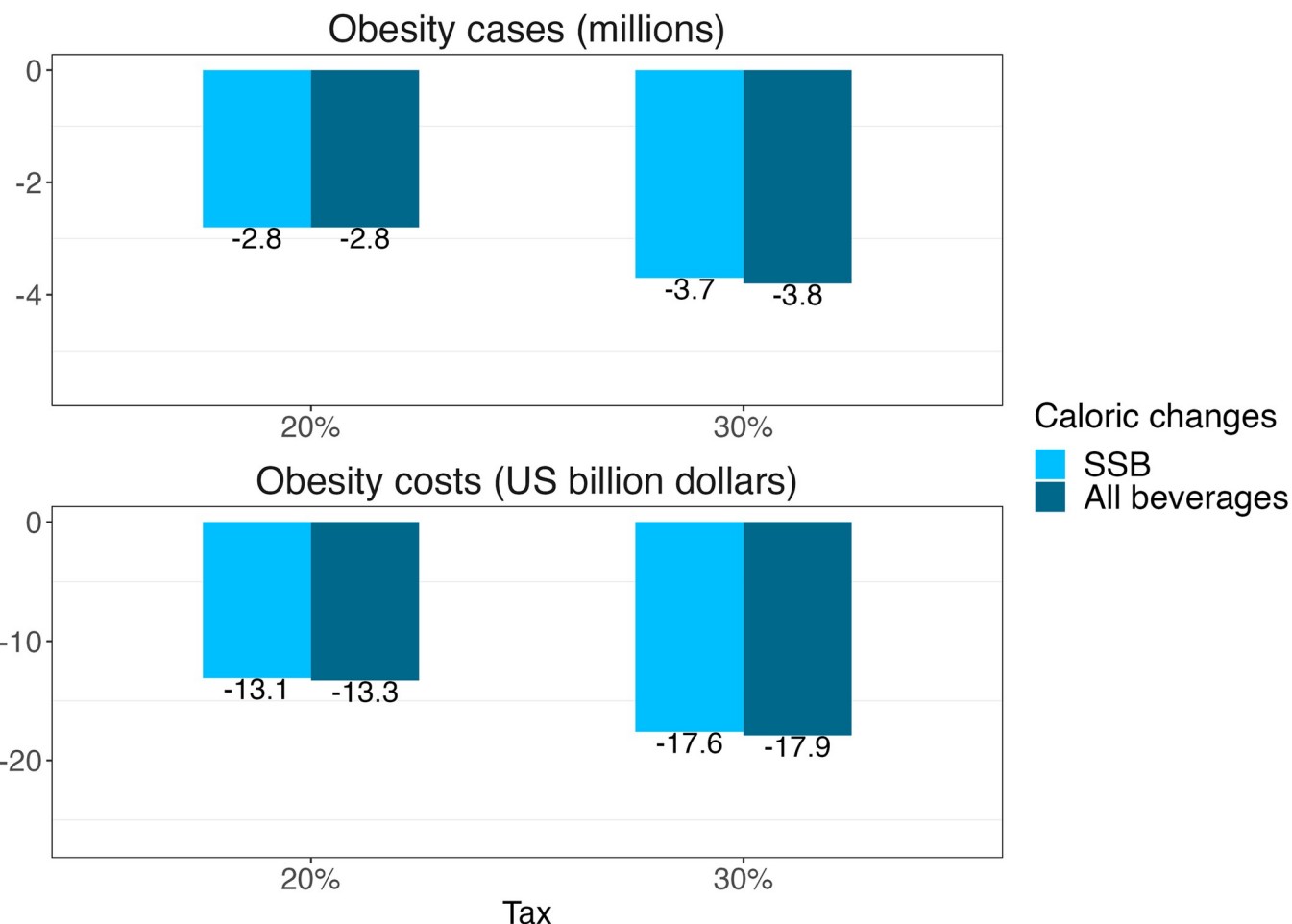

**Fig 2. Reduction in cases and cumulative cost savings expressed in 2021 US$ 10 years after implementing an SSB tax, using own-price elasticities considering the effect on sugar-sweetened beverages and using cross-price elasticities considering all beverages.** SSB, sugar-sweetened beverage.

that alcoholic beverage purchases are expected to decrease if SSB prices increase, while non-caloric beverages purchases will increase only among adults in the high-income group. Considering these substitutions, a 20% tax should lead to a direct caloric reduction of 16.9 kcal from SSB, which will be further increased by substitution or complementation for other liquids with a lower caloric density (−0.4 kcal after a 20% tax and −0.6 kcal after a 30% tax). This finding implies that if an SSB tax is implemented, no caloric substitution for liquids is to be expected; an observation that is in line with a prior observational study that found a weak caloric compensation for SSB in the Brazilian population [36] and with international studies that suggest that caloric compensation for liquids is incomplete [37].

While the SSB tax is being discussed in Brazil, information about the potential impact of a 20% and 30% SSB tax on obesity is limited. Based on our modeling exercise, a 20% tax could result in a 6.7% reduction in obesity, increasing to 9.1% under a 30% tax. This estimate is higher than what was previously estimated in Brazil, in which a 20% tax was predicted to increase obesity by 1.3% in men and by 2.4% in women [38]. Differences in estimates are likely due to differences in the method of estimating own- and cross-price elasticities and the caloric densities used. We predicted a 17.3 kcal/day/person reduction, considering own- and cross-price elasticities, yet the study by Enes and colleagues [38] predicted an increase of 8 kcal/day/

person, also considering own- and cross-price elasticities. Both estimates were generated using the same dataset (POF, 2017–2018), yet, Enes and colleagues [38], used censored equations at the household level, while we estimated elasticities at the sample unit level, to avoid censored data. Differences in the categorization of food and beverage items and in the caloric content of products could have also contributed to the reported divergences. Given the observed difference, we searched for independent assessments of elasticities in Brazil and found a recent report by Venson and colleagues [39] using the same dataset (the 2017–2018 Brazilian Household Budget Survey), a similar model (QUAIDS model) together with censored demand and household data [39]. Using the elasticities by Venson and colleagues, we found similar results to our analysis, suggesting that this more sophisticated approach concurs with our analysis. Our approach allowed us to estimate elasticities and effects by income tertile, an important condition to assess the expected impacts by socioeconomic status. Also, this estimate is within the range of prior modeling studies using a comparable approach. For instance, in Mexico, a 20% tax was estimated to reduce 16.7 kcal using yearly estimate and 31.1 kcal using a monthly estimate [9].

We observed important differences in the expected impact on obesity rates by income. High-income groups are expected to have their obesity prevalence reduced by 7.8% compared with 5.4% in low-income groups. This could be explained by differences in baseline consumption of SSB, which were 86.7 kcal/day for high-, 74.0 kcal/day for middle-, and 53.5 kcal/day for low-income groups. Also, baseline obesity prevalence was 27.0% in the high-, 26.2% in the middle-, and 22.3% in the low-income groups. This, along with greater baseline SSB consumption, could explain the greater expected reductions in BMI among high-income individuals, despite having lower price-elasticities in comparison with low-income individuals. We found that a 20% and 30% excise tax on SSB will produce benefits for all income groups as it will reduce obesity levels in all SES groups; however, an investment of tax revenue could further benefit lower-income groups if used to fund redistributive social policies [40].

An SSB tax in Brazil is expected to produce important savings related to obesity prevention. We estimated a reduction of US$ 13.3 billion in obesity costs 10 years after the implementation of a 20% SSB tax (US$ 5.6 billion including only direct costs, available in appendix). The US $5.6 billion reduction is smaller than what has been estimated in the US (US$ 23.6 billion) but higher than for Mexico (US$ 159.5 million) for a similar tax increase (20%) and a similar period (10 years) [12,41]. We compared obesity direct costs savings only, because these 2 previous studies only measured direct obesity costs. The differences across countries could be due to differential obesity costs per case, which vary by how healthcare is financed and organized and by differences in obesity prevalence and estimated reductions. Finally, our cost-analysis does not consider the potential benefits derived from the social investment of the tax revenue, particularly if revenue is directed at strengthening obesity prevention measures, which can further increase the health benefits from the tax.

Some limitations of our study should be mentioned. Baseline weight and height were self-reported. We relied on a quantile distribution method to calibrate it using objectively measured weight and height available at the Brazilian National Health Survey (Section 2 in S1 Appendix). Yet, this method is imperfect and could be insufficient to correct reporting bias [30]. For that reason, we performed a sensitivity analysis using the same dynamic model, but at an aggregated level. In this analysis, we used objectively measured weight and height from 14 groups (aggregated data) and linked it to SSB consumption. In the 14 groups, the combined estimation fell within the confidence intervals for the individual model (Table H in S1 Appendix). This sensitivity analysis increases our confidence in the correction we implemented on the self-reported measures to estimate weight reductions in kilograms. A similar approach could not be used to assess the effect of the correction on the BMI classification. However, we estimated a 16.9% obesity prevalence using self-reported weight and height, compared to 25.2% using adjusted self-

reported data, and to 26.8% obesity prevalence using objective measures of weight and height (Table B in S1 Appendix). The potential error of adjusting self-reported data on the BMI classification (obesity versus non-obesity) in comparison with using measured data is in the downward direction, suggesting that adjusted data could underestimate the SSB impact on obesity, but in a lower magnitude if we would have used self-reported data without adjustments.

Our model could overestimate cost savings. Prevalent costs of obesity were obtained from Okunogbe and colleagues [24], yet, the obesity changes estimated in our model will take years to fully materialize after the intervention. Some indicators will improve faster after weight loss, such as disease incidence, but others will take years to decades to change, such as mortality. To fully model these dynamics, we would have needed to use a life-table model with an open population to capture these changes, discount the health gains over time, and go beyond the 10-year modeling timeframe used in our analysis. We attempted to reduce overestimation by introducing a 3-year lag of no cost savings and including a 5% discount rate, but this could still fall short. Future studies interested in modeling obesity cost savings should rely on life-table models to capture the net present value of health gains. Also, the literature on obesity costs in Brazil provides heterogeneous figures. A systematic review found that obesity costs in Brazil ranged from USD 133.8 million to USD 6.3 billion per year (47 times the lowest estimation) [42]. We relied on an international estimation that includes 8 countries, uses a societal perspective, and found a higher obesity cost compared to the highest estimation included in the systematic review. We are not considering that over the 10 years simulated, the Brazilian population will age, which could affect the average cost of obesity; our cost per person could underestimate the cost at the end of the simulation period. Our estimates depend on a steady-state assumption for body weight, implying a constant obesity prevalence over time; however, the prevalence of obesity in Brazil has increased over time (11.8% to 20.3% between 2009 and 2019). The impact of this assumption depends on the drivers of the obesity trend in Brazil, with the worst-case scenario being an overestimation of the effect that could occur if obesity increases are driven by products not included in the SSB tax. However, SSB are recognized as important drivers of the obesity epidemic in the world. Despite this, future studies should aim at including obesity trends into models. Finally, we simulated this intervention in the adult population, so further reductions in obesity prevalence are to be expected in children and adolescents who may change dietary habits and reduce weight as they transition into adulthood.

SSB taxes have been implemented in more than 73 countries worldwide [7]. Studies have shown that SSB are elastic and respond to increases in price; yet no negative economic effects have been observed in countries where SSB taxes have been implemented [40], and analyses in Brazil suggest an economic net benefit from implementing an SSB tax [43]. Our findings suggest that implementing an SSB tax in Brazil will lead to an important reduction in the caloric intake of beverages with little to no nutritional added value; this reduction could help reduce the obesity prevalence across all socioeconomic strata. From an economic perspective, an SSB tax could reduce costs while generating revenue that could be in turn invested in other obesity prevention interventions. In 2016, the WHO recommended that all countries should implement an SSB tax of at least 20% [44]. Our analysis suggests that a 30% tax in Brazil could produce even greater population health benefits for all income strata.

## Supporting information

**S1 Appendix. Additional information on the simulation model, parameters, assumptions, and supplementary results.**
(PDF)

## Author Contributions

**Conceptualization:** Tonatiuh Barrientos-Gutierrez, Ana Clara Duran.

**Data curation:** Ana Basto-Abreu, Rossana Torres-Alvarez, Paula Pereda, Ana Clara Duran.

**Formal analysis:** Ana Basto-Abreu, Rossana Torres-Alvarez, Tonatiuh Barrientos-Gutierrez, Paula Pereda.

**Funding acquisition:** Tonatiuh Barrientos-Gutierrez, Ana Clara Duran.

**Investigation:** Rossana Torres-Alvarez, Tonatiuh Barrientos-Gutierrez, Ana Clara Duran.

**Methodology:** Ana Basto-Abreu, Rossana Torres-Alvarez, Tonatiuh Barrientos-Gutierrez, Paula Pereda, Ana Clara Duran.

**Project administration:** Ana Clara Duran.

**Supervision:** Tonatiuh Barrientos-Gutierrez.

**Writing – original draft:** Ana Basto-Abreu, Rossana Torres-Alvarez, Tonatiuh Barrientos-Gutierrez, Paula Pereda, Ana Clara Duran.

**Writing – review & editing:** Ana Basto-Abreu, Rossana Torres-Alvarez, Tonatiuh Barrientos-Gutierrez, Paula Pereda, Ana Clara Duran.

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
