## [Editor Report · Decision Letter 0]

10 Oct 2023

Dear Dr Barrientos-Gutierrez, 

Thank you for submitting your manuscript entitled "Expected reduction in obesity of a 20% and 30% tax to Sugar-Sweetened-Beverages in Brazil: a modeling study" for consideration by PLOS Medicine.

Your manuscript has now been evaluated by the PLOS Medicine editorial staff as well as by an academic editor with relevant expertise and I am writing to let you know that we would like to send your submission out for external peer review.

Please re-submit your manuscript within two working days, i.e. by Oct 12 2023 11:59PM.

Kind regards,

Katrien G. Janin, PhD

Senior Editor

PLOS Medicine

---

## [Decision Letter · Decision Letter 1]

6 Nov 2023

Dear Dr. Barrientos-Gutierrez,

Thank you very much for submitting your manuscript "Expected reduction in obesity of a 20% and 30% tax to Sugar-Sweetened-Beverages in Brazil: a modeling study" (PMEDICINE-D-23-02904R1) for consideration at PLOS Medicine. 

[LINK]

In light of these reviews, I am pleased to tell you that we would like to consider a revised version that addresses the reviewers' and editors' comments. Obviously we cannot make any decision about publication until we have seen the revised manuscript and your response, and we plan to seek re-review by one or more of the reviewers. 

We expect to receive your revised manuscript by Nov 27 2023 11:59PM. Please email us (plosmedicine@plos.org) if you have any questions or concerns.

We look forward to receiving your revised manuscript. 

Best wishes,

Philippa Dodd - PLOS Medicine, on behalf of:

Katrien Janin, PhD

PLOS Medicine

plosmedicine.org

COMMENTS FROM THE ACADEMIC EDITOR

I'm supportive of the paper, but have a few comments.

1. I don't see any particular rationale for essentially replicating similar studies that have been conducted elsewhere beyond no-one's done it in Brazil yet. It would be helpful to understand whether the authors expect price elasticity of SSBs in Brazil to be different from elsewhere and why. I guess really my underlying question is why are modelling studies from elsewhere not generalisable to Brazil? I think there are probably some good reasons, but it would be good to be explicit.

2. I don't understand how NOVA would be used to classify beverages into the four categories listed - NOVA categorises based on degree of industrial processing.

3. Consumption was based on individual data for participants aged 20y+. In many countries SSB consumption peaks in late childhood. So I expect that their estimates of population consumption are underestimates. This could be considered further.

COMMENTS FROM THE EDITORS

GENERAL: 

For in-text reference, citations are placed within square parentheses and should precede punctuation. (e.g see line 83: ‘… metabolic syndrome, and cancer.[1–3] Governments … ‘ should be changed to ‘ ‘…metabolic syndrome, and cancer [1–3].’‘ 

Please amend throughout.

Where you provide CI values, please also provide p values for all results where appropriate (e.g see Table 2 and 3), check and amend throughout. For p values, please report as p<0.001 and where higher as 'p=0.002'. Please add the statistical method used to your method section. 

We also invite you to report p values to consistently to the third decimal digit - thousandths (e.g see table Table 1)

MODELLING STUDIES 

Of all authors who submit a modelling study we ask for inclusion of specific items, derived from Geoffrey P Garnett, Simon Cousens, Timothy B Hallett, Richard Steketee, Neff Walker. Mathematical models in the evaluation of health programmes. (2011) Lancet DOI:10.1016/S0140-6736(10)61505-X.

Please ensure all the items listed below are included with your manuscript. Please review the list below and confirm/revise as necessary:

i) Please provide a diagram that shows the model structure, including how the disease natural history is represented, the process and determinants of disease acquisition, and how the putative intervention could affect the system.

ii) Please provide a complete list of model parameters, including clear and precise descriptions of each parameter, together with the values or ranges for each, with justification or the primary source cited, and important caveats about the use of these values noted.

iii) Please provide a clear statement about how the model was fitted to the data [including goodness-of-fit measure, the numerical algorithm used, which parameter varied, constraints imposed on parameter values, and starting conditions].

iv) For uncertainty analyses, please state the sources of uncertainties quantified and not quantified [can include parameter, data, and model structure].

v) Please provide sensitivity analyses to identify which parameter values are most important in the model. Uncertainty estimates seek to derive a range of credible results on the basis of an exploration of the range of reasonable parameter values. The choice of method should be presented and justified.

vi) Please discuss the scientific rationale for this choice of model structure and identify points where this choice could influence conclusions drawn. Please also describe the strength of the scientific basis underlying the key model assumptions.

DATA AVAILABILITY:

Thank you for making all the data fully available without restriction. Given that your study is a modelling study, you are also responsible for providing the source code needed to replicate the study's findings in a repository (such as GitHub, SourceForge or Bitbucket) or a cloud computing service (such as Code Ocean). Please explain in the Data Availability Statement how readers can access or will be able to access the shared code. 

ABSTRACT:

In the last sentence of the Abstract Methods and Findings section, please describe the main limitation(s) of the study's methodology.

AUTHORS SUMMARY:

Ideally each sub-heading should contain 2-3 single sentence, concise bullet points containing the most salient points from your study.

In the final bullet point of ‘What Do These Findings Mean?’ Please include the main limitations of the study in non-technical language.

DISCUSSION:

In line 338: Please avoid assertions of primacy. We suggest you add ‘ to our knowledge’ or something alike. 

In line 398 the term "trend" is used. The term trend should be used only when the test for trend has been conducted. Please revise accordingly. 

REFERENCES:

As mentioned above, for in-text reference, citations are placed within square parentheses and should precede punctuation. Please amend throughout. 

For online references (e.g see reference 1), please use the following format: [accessed on 2 Apr 2021] instead of [cited 2 Apr 2021].

FIGURES and TABLES:

Please ensure you re-define abbreviations in the legends of your figures and tables (e.g Figure 1, Table 1, 2 etc.). 

COMMENTS FROM THE REVIEWERS:

Reviewer #1: Dear authors, this is an important study, which aimed to estimate the price elasticity of sweetened drinks and other drinks and evaluate the impact of taxation on sweetened drinks on the prevalence of obesity in Brazil. There are few studies that have evaluated the impact of fiscal measures to reduce SSB consumption on health outcomes, especially in low- and middle-income countries. The study makes important contributions to the scientific community in the area of impact assessment of tax policies.

Title: I suggest that the aspect of impact on cost be incorporated into the title as this was an important objective of the study.

Page 3 line 86: I suggest removing the word recent, as this is a study published in 2019.

Introduction:

In 2022, Brazil approved a bill that is still under analysis by the economic affairs committee that provides for a 20% tax on sugary drinks. I think it is important to include this information in the introduction to inform the reader where the country is in this fiscal policy.

The authors should make it clearer in the introduction about the importance of the analysis carried out in this study, that is, why it is relevant to estimate the potential impact of SSB taxation on both obesity and costs in the health system. Other studies in this regard have already been published in Brazil. It is important to include. Enes, C.C.; Rinaldi, A.E.M.; Nucci, L.B.; Itria, A. The Potential Impact of Different Taxation Scenarios towards Sugar-Sweetened Beverages on Overweight and Obesity in Brazil: A Modeling Study. Nutrients 2022, 14, 5163. https://doi.org/10.3390/nu14235163

Methods

Why were weight and height data not used directly from PNS 2019, since POF 2017-2018, from which consumption data were obtained, was carried out in a period close to PNS 2019? Although it is not possible to combine the databases at the individual level, when establishing the study population >= 20 years old, it is possible to cross-reference consumption data with obesity prevalence data, as the research was carried out at similar times. This way there would be no need to calculate adjustments for weight and height.

The dynamic model proposed by Hall et al predicts that 50% of the expected weight loss will occur in the first year after taxation and 95% of the reduction by the end of the third year of taxation. How was this considered in the model to calculate weight reduction over a 10-year period?

Results

Page 7 line 272: remove the indication of % -25.3% as it is kcal

Discussion

In line 338, the authors state that this is the first study in Brazil that estimated the impact of SSB taxation on body weight. There is a mistake in this statement because a study was published in 2022 that also proposed to evaluate the impact of SSB taxation. Enes, C.C.; Rinaldi, A.E.M.; Nucci, L.B.; Itria, A. The Potential Impact of Different Taxation Scenarios towards Sugar-Sweetened Beverages on Overweight and Obesity in Brazil: A Modeling Study. Nutrients 2022, 14, 5163. https://doi.org/10.3390/nu14235163

Even the results found in this study were different from those presented in this article and the possible reasons for these differences should be discussed and explored

Reviewer #2: This is a well-written paper on a topic that is of interest to a wide health readership. The methods are sound and the findings are novel. The conclusions are supported by the results. The results are immediately relevant to Brazil, and also of value to similar countries in the region and around the world. 

The derivation of price elasticities, by income tertile, using Brazilian data, is a clear strength of this work. So are the use of a microsimulation model with Hall et al's energy balance, and the correction applied to self-reported height and weight, the latter with an alternative approach in the sensitivity analysis for validation purposes.

I have a few comments, but none detract from the great work you have done for this paper.

The approach to estimating health care cost is suitable for ballpark estimates but is likely to underestimate the true health care cost savings by a considerable margin, as the authors duly indicate in their discussion. Not mentioned is that the cost per case is the same across ages, whereas in reality it would rise with age as the prevalence of obesity-related conditions rises. Given population aging, not factoring this in would lead for further underestimation of the cost savings.

Line 249 / 394 / S1 appendix section 5: I believe that usually, the health system perspective does include out-of-pocket expenses. The perspective used is perhaps better described as a 'payer perspective', as per an ISPOR Special Task Force Report (Garrison et al, ViH 2018; https://doi.org/10.1016/j.jval.2017.12.006) (although in that guidance the term 'health care sector perspective' is used). 

Consider reporting on the results of sensitivity analysis in results section, even if it's in one sentence.

Line 411: Add a reference for the statement that "In 2017, the World Health Organization recommended that all countries should implement an SSB tax of at least 20%."

S1, Fig. B: The adjustments made to height seem to have exacerbated the clustering of values around round numbers, no doubt due to the splines. If there were a way to smooth the adjusted curves, that wo

---

## [Decision Letter · Decision Letter 2]

16 Feb 2024

Dear Dr. Barrientos-Gutierrez,

Thank you very much for re-submitting your manuscript "Expected reduction in obesity prevalence and costs of a 20% and 30% ad valorem excise tax to Sugar-Sweetened-Beverages in Brazil: a modeling study" (PMEDICINE-D-23-02904R2) for review by PLOS Medicine.

I have discussed the paper with my colleagues and the academic editor and it was also seen again by the reviewers. I am pleased to say that provided the remaining comments of reviewer #2 are dealt with we are planning to accept the paper for publication in the journal.

[LINK]

If you have any questions in the meantime, please contact me (kjanin@plos.org) or the journal staff (plosmedicine@plos.org).  

We look forward to receiving the revised manuscript by Feb 23 2024 11:59PM.   

Sincerely,

Katrien Janin, PhD

Senior Editor 

PLOS Medicine

plosmedicine.org

Requests from Editors:

Comments from Reviewers:

Reviewer #1: Dear editor, I thank you for the opportunity to evaluate the second version of the article. Analyzing the document sent by the authors, I realized that the notes I presented were responded to appropriately, and justifications were also presented. Furthermore, considering the notes from the other reviewers, I see that the new version of the article provides more complete information. I have no further notes on the present.

Reviewer #2: The paper has improved, particularly, the perspective and costs are clearer. It did occur to me that there is an issue with the timing of the health care cost and productivity savings that I missed in my previous reading of the paper.

The source study, Okunogbe et al, estimated those costs using a cost-of-illness approach, which aims to take a snapshot of costs in a particular year. Some of these costs are the result of exposures in prior years. The most extreme example is the mortality component of productivity losses, which are calculated by attributing a proportion of deaths to obesity, then multiplying with (productive) life expectancy at the age of death, and with wages. In the model in the present paper, all costs are related to obesity in the year of analysis.

In reality, a change in SSB consumption would translate to a change in body mass (this is covered very well in the model), which would reduce the number of new cases of obesity-related disease (diabetes, heart disease, etc.). Over time, the pool of prevalent cases prevented would rise. Impacts on mortality would follow, and mortality-related productivity gains (and health care costs in added years of life) would trail those. The model used here omits several of these steps, so part of the cost savings are brought forward, potentially by up to decades.

This could be incorporated in the analysis if the approximate timing of the cost savings and productivity gains can be estimated, and the model can be refined to deal with those lagged effects. Alternatively, this could be addressed in the discussion section.

(That does not solve the issue around 'health care costs in added years of life' - see e.g. Van Baal et al, Health Econ. 2007 Apr;16(4):421-33 - which is ignored in cost-of-illness studies too, and best left to future studies that use lifetable modelling.)

Reviewer #3: Thank you for your close attention to the comments given by the editors and all four reviewers. I have no additional suggestions for revision at this time. 

Reviewer #4: In my opinion, the authors have adequately addressed questions and concerns raised by me and other reviewers. I think the paper is acceptable in its current form. Congratulations to the authors!

[LINK]

---

## [Decision Letter · Decision Letter 3]

14 Mar 2024

Dear Dr. Barrientos-Gutierrez,

Thank you very much for re-submitting your manuscript "Expected reduction in obesity prevalence and costs of a 20% and 30% ad valorem excise tax to Sugar-Sweetened-Beverages in Brazil: a modeling study" (PMEDICINE-D-23-02904R3) for review by PLOS Medicine.

I have discussed the paper with my colleagues and the academic editor and it was also seen again by the reviewer. I am pleased to say that provided the remaining editorial and production issues are dealt with we are planning to accept the paper for publication in the journal.

[LINK]

We look forward to receiving the revised manuscript by Mar 21 2024 11:59PM.   

Sincerely,

Katrien Janin, PhD

Senior Editor 

PLOS Medicine

plosmedicine.org

Comments from Reviewers:

Reviewer #2: The work now contains a lag time between the start of the intervention, and the costs. And a 5% discount rate has been applied. Both are improvements, but the chosen solutions still falls a bit short, I'm afraid.

In reference 30, the BODE3 team wrote, in Table 1: "Time lags for intervention effect: It takes time for a change in body mass index (BMI) to impact on disease incidence. As there are no precise data on just how long these are, we have used wide windows of time lags. For cancers, the time lag is assumed to range between 10 and 30 years. For CHD, stroke, diabetes, and osteoarthritis (the noncancers), the time lag is assumed to be shorter and ranges between 0 and 5 years. Wide uncertainty is included around these estimates."

That is, the lag time accounts for the impact of a risk factor change on incidence. Cleghorn et al then use a proportional multi-state life table model that accounts for the subsequent lags between incidence and prevalence, and prevalence and mortality, and for the fact that some lost life years are further into the future than others. 

To illustrate, I had hoped to add a few graphs. The system will not allow that, but I'll see if the editorial team can help. The graphs show the results of a study in which we modelled the potential impact of an SSB tax (compared to BAU) using a model similar to the one used by BODE3. The graphs show the time patterns in IHD, where impact is quick for incidence (we modelled no lag time between exposure change and incidence), slower for prevalence (this builds up over time), and mortality (which lags more as it is a function of prevalence), and HALYs (not visible but this lags even more as it is partly a function of deaths in previous years).

In this paper, you now apply the delay factors intended for incidence, to all those sequelae. (As explained in my previous review, the Okunogbe paper used a cost-of-illness approach, which takes a snapshot of costs in a particular year.) In years 4-10, you use the full impact, which still underestimates the lag. Most of the impact would materialise after that 10-year mark. It would nevertheless be fair to attribute those cost savings in later years to the exposure changes within those first 10 years - it's just that they would properly need to be discounted in a lifetable or similar structure. Since the benefits would stretch over decades, the impact of discounting would be much greater than in your current calculations, which go up to year 10. In another graph below, you can see that with a 5% discount rate, the 'net present value' for an event in year 10 is 61%, but a life year in year 25 would count only for 0.3 life years (NPV).

This would affect mainly the cost of premature mortality, which in Okunogbe's study "constitute[d] a substantial proportion of indirect costs (about 56%-92%) across all countries". Although Okunogbe et al ignore this, it would apply also, to a lesser extent, to the direct obesity costs. Longstanding obesity is more costly than recent obesity due to the time it takes to develop disease (see above).

Modelling this properly is clearly not an option here - you would need a sophisticated life table model. The current solution is not elegant and still overestimates the 10-year gains, and this is not explained in the paper; not mentioned in the limitations. 

The best solution would be to omit the estimates of the cost savings altogether, and instead focus on the reductions in obesity (by income), which is where the strength of this paper lies. The discussion could then mention the likely impact on obesity-related costs, referring to the Okunogbe paper.

Alternatively, you could proceed as you did now, but with more discussion in the limitations section. You would have to point out the cost savings could be attributed to changes in obesity prevalence in those 10 years, but would in reality materialise largely in the decades following, that discounting would reduce the net present value of the gains (which you accounted for only partially), and that future life table-based modelling could produce more realistic estimates of these impacts over time.

Minor comment: 'dietary multistate life table' rather than 'Diet multistage life table'.

Requests from Editors:

After a long discussion with my editorial colleagues, and taken fully on board the reviewer comments, we like to suggest the following solution to move forward: 

We prefer to go down the route to ask you to expanded the limitations in the discussion and ask to use very appropriate language when talking about the projected cost savings in your manuscript. The reason we prefer to go down that route is that the current calculation do give an indication of the cost involved but we agree with the reviewer's points about time lag and the resulting likelihood that their cost savings estimates are inflated MUST be discussed carefully in the discussion/limitations. We therefore ask you to implement the alternative option as presented by the reviewer.

We like to thank the reviewer for such a terrific and informative discussion

Please do not hesitate to contact me directly at kjanin@plos.org if you have any question about the above.

[LINK]

---

## [Decision Letter · Decision Letter 4]

5 Apr 2024

Dear Dr Barrientos-Gutierrez, 

On behalf of my colleagues and the Academic Editor, [AE Name], I am pleased to inform you that we have agreed to publish your manuscript "Expected reduction in obesity prevalence and costs of a 20% and 30% ad valorem excise tax to Sugar-Sweetened-Beverages in Brazil: a modeling study" (PMEDICINE-D-23-02904R4) in PLOS Medicine.

I do have one editorial request: please change your title to "Estimated reduction in obesity prevalence and costs of a 20% and 30% ad valorem excise tax to Sugar-Sweetened-Beverages in Brazil: a modeling study".

PRESS

Thank you again for submitting to PLOS Medicine. We look forward to publishing your paper!

Sincerely, 

Katrien G. Janin, PhD 

Senior Editor 

PLOS Medicine